

# Australian vegetation phenology: new insights from satellite remote sensing and digital repeat photography

Caitlin E. Moore[1], Tim Brown[2], Trevor F. Keenan[3,4], Remko A. Duursma[5], Albert I.J.M. van Dijk[6], Jason Beringer[7], Darius Culvenor[8], Bradley Evans[9,10], Alfredo Huete[11], Lindsay B. Hutley[12], Stefan Maier[13], Natalia Restrepo-Coupe[11], Oliver Sonnentag[14], Alison Specht[15], Jeffrey R. Taylor[16], Eva van Gorsel[17], Michael J. Liddell[18]

[1] School of Earth, Atmosphere and Environment, Monash University, Clayton, VIC, 3800, Australia.

[2] Research School of Biology, Plant Sciences, Australian National University, Acton, ACT, 0200 Australia.

[3] Department of Biological Sciences, Macquarie University, North Ryde NSW 2109, Australia.

[4] Lawrence Berkeley National Lab., 1 Cyclotron Road, Berkeley, CA, 94720, USA

[5] Hawkesbury Institute for the Environment, University of Western Sydney, Locked Bag 1797, Penrith, NSW, 2751, Australia

[6] Fenner School of Environment & Society, The Australian National University, Acton, ACT, 2601, Australia.

[7] School of Earth and Environment, University of Western Australia, Crawley 6009, WA, Understory Australia.

[8] Environmental Sensing Systems, 16 Mawby Road, Bentleigh East, VIC, 3165, Australia.

[9] Department of Environmental Sciences, The University of Sydney, Eveleigh, NSW, 2015, Australia.

[10] Terrestrial Ecosystem Research Network Ecosystem Modelling and Scaling Infrastructure, The University of Sydney, Eveleigh, NSW, Australia, 2015.

[11] Plant Functional Biology and Climate Change Cluster, University of Technology Sydney, Broadway, NSW, 2007, Australia.

[12] School of Environment, Research Institute for the Environment and Livelihoods, Charles Darwin University, Casuarina, NT, Australia, 0909

[13] maitec, P.O. Box U19, Charles Darwin University, Darwin, NT, 0815, Australia.

[14] Département de Géographie, Université de Montréal, Montréal, Québec, H3C 3J7, Canada.

[15] Australian Centre for Ecological Analysis and Synthesis, The University of Queensland, Brisbane, QLD, 4072, Australia.

[16] Institute of Arctic and Alpine Research, University of Colorado, Boulder, CO, 80301, USA.

[17] CSIRO, Ocean and Atmosphere Flagship, Yarralumla, ACT, 2601, Australia.

[18] College of Science, Technology and Engineering, James Cook University, Cairns, QLD, 4878; Australian Supersite Network, Terrestrial Ecosystem Research Network.

*Correspondence to:* Caitlin E Moore (caitlin@moorescience.com.au)



**Abstract**
Phenology is the study of periodic biological occurrences and can provide important insights into the
influence of climatic variability and change on ecosystems. Understanding Australia's vegetation
phenology is a challenge due to its diverse range of ecosystems, from savannas and tropical rainforests
to temperate eucalypt woodlands, semi-arid scrublands and alpine grasslands. These ecosystems exhibit
marked differences in seasonal patterns of canopy development and plant life-cycle events, much of
which deviates from the predictable seasonal phenological pulse of temperate deciduous and boreal
biomes. Many Australian ecosystems are subject to irregular events (i.e., drought, flooding, cyclones
and fire) that can alter ecosystem composition, structure and functioning just as much as seasonal
change. We show how satellite remote sensing and ground-based digital repeat photography (i.e.
phenocams) can be used to improve understanding of phenology in Australian ecosystems. First, we
examine temporal variation in phenology at the continental scale using the Enhanced Vegetation Index
(EVI), calculated from MODerate resolution Imaging Spectroradiomter (MODIS) data. Spatial
gradients are revealed, ranging from regions with pronounced seasonality in canopy development (i.e.,
tropical savannas) to regions where seasonal variation is minimal (i.e., tropical rainforests) or high but
irregular (i.e., arid ecosystems). Next, we use time series colour information extracted from phenocam
imagery to illustrate a range of phenological signals in four contrasting Australian ecosystems. These
include greening and senescing events in tropical savannas and temperate eucalypt understory, as well
as strong seasonal dynamics of individual trees in a seemingly static evergreen rainforest. We also
demonstrate how phenology links with ecosystem gross primary productivity (from eddy covariance)
and discuss why these processes are linked in some ecosystems but not others. We conclude that
phenocams have the potential to greatly improve current understanding of Australian ecosystems. To
facilitate sharing of this information, we have formed the Australian Phenocam Network
(http://phenocam.org.au/).
**Key Words**
Phenocam, OzFlux, remote sensing, climate change, MODIS, EVI
**1 Introduction**
Phenology is the study of the timing of periodic biological occurrences, from flowering and leaf
emergence to animal migrations and breeding patterns, all of which vary between and within species
and are influenced to a certain degree by the climate (Lieth, 1974;Richardson et al., 2013a).
Phenological monitoring allows us to quantify and track seasonal cycles and irregular natural events, as
well as how vegetation phenology responds to meteorological events and climate change. At the
ecosystem level, the timing of leaf emergence and senescence in plants has a major influence on the
length of the growing season (Dragoni et al., 2011;Keenan et al., 2014b). At larger scales, phenological
responses feed back to the climate system by altering the land surface energy balance and the coupled
cycles of water, carbon and nutrients from an ecosystem (Peñuelas et al., 2009;Richardson et al., 2013a).





Research suggests that contemporary warming has advanced spring onset in the northern hemisphere
by *ca.* 2.8 days per decade since the 1970's (Parmesan, 2007), with even higher rates of change (4.8
days per decade) in more recent decades (Keenan et al., 2014b). Likewise, in the southern hemisphere,
advancement in spring onset has been estimated at a rate of 2.2 days per decade (Chambers et al., 2013).
In response to increasing evidence of earlier spring onsets, the Intergovernmental Panel on Climate
Change (IPCC) has placed a high importance on the gathering and interpretation of phenology
observations to improve our understanding of the ecological impacts of climate change (Ciais et al.,
2013). Understanding the processes that lead to modifications in phenology, as well as the complex
links between phenological shifts and climatic variability and change at local, regional and global
scales, requires more detailed field-based observations that underpin larger scale remote sensing and
ecological modelling tools (de Jong et al., 2012;Migliavacca et al., 2012;Melaas et al., 2013).
Repeat photography with digital cameras (phenocams) has emerged as a powerful observational tool
for ecological research and evaluation of remote sensing data products and model simulations
(Richardson et al., 2007;Migliavacca et al., 2011;Hufkens et al., 2012;Wingate et al., 2015;Brown et
al., 2016). Digital images collected from phenocams typically provide red-green-blue (RGB) colour
channel information that can be partitioned and converted to quantitative indices (Woebbecke et al.,
1995;Gillespie et al., 1987;Sonnentag et al., 2012). These indices have been shown to closely track
vegetation colour changes, such as canopy greenness, and are indicative of different plant development
stages in a variety of ecosystems (Sonnentag et al., 2011;Inoue et al., 2014;Keenan et al., 2014a;Peichl
et al., 2014;Toomey et al., 2015). A loosely organized global phenocam network is emerging with
regional networks including the PhenoCam Network (Richardson et al., 2009a) and the National
Ecological Observatory Network (NEON) in the USA (Keller et al., 2008), the EUROPhen network in
Europe (Wingate et al., 2015), the Phenological Eyes Network (PEN) in Japan (Nasahara and Nagai,
2015) and most recently, the Australian Phenocam Network (Brown et al, 2016). These networks
provide support for phenocam data collection and sharing in the wider scientific community and are a
platform for the development of open data and data sharing standards.
Phenology information can also be derived from satellite remote sensing imagery. Satellite data
provides spatially explicit time series of vegetation phenology proxies such as the enhanced vegetation
index (EVI, Huete et al., 2002), the normalised difference vegetation index (NDVI, Tucker, 1979), as
well as estimates of canopy cover fraction, the fraction of absorbed photosynthetically active radiation
and leaf area index (LAI) derived from optical remote sensing (Gonsamo and Chen, 2014). In addition,
vegetation optical depth and biomass can be derived from radar or passive microwave remote sensing
(Andela et al., 2013;Liu et al., 2013;Liu et al., 2011;Liu et al., 2007). Such remotely sensed vegetation
phenology proxies can be analysed to estimate temporal changes in the vegetation characteristics at the
landscape scale and contribute toward the development of satellite phenology products, such as the



MODerate resolution Imaging Spectroradiometer (MODIS) Land Cover Dynamics product (Ganguly
et al., 2010), and the Australian satellite land surface phenology product (Broich et al., 2015).
Satellite-derived phenology can be compared to changes observed at smaller scales with phenocams.
For example, Hufkens et al. (2012) and Klosterman et al. (2014) used phenocams to evaluate the
MODIS MCD12Q2 product for deciduous forests in the north-eastern USA. Despite discrepancies
related to spatial scale and site representativeness, the results showed good agreement between the
phenocam- and satellite-derived timing of phenological events (i.e. phenophases). For various
temperate and boreal forest, grassland and peatland ecosystems, time-series data of canopy greenness
derived from phenocam imagery have been used to analyse seasonal changes of net ecosystem exchange
(NEE) and gross primary productivity (GPP) (Ahrends et al., 2009;Richardson et al., 2007;2009;Zhu et
al., 2013;Toomey et al., 2015). Characterising changes in plant physiology (Keenan et al., 2014a),
modelling of canopy development (Wingate et al 2015) and quantifying inter-annual variation in
phenology (Inoue et al 2014) have also been achieved using phenocams. Despite the promising
applications of phenocam data in North America, Europe and Asia, there has been a slow uptake of
phenocam research in Australia; a continent characterized by diverse ecosystem types often with
distinctly different phenological characteristics compared to temperate and boreal deciduous
ecosystems (Chambers et al., 2013).
The diversity of Australian ecosystems, and their marked differences to northern hemisphere temperate
deciduous and boreal biomes, poses the challenge of how to best define and quantify vegetation
phenology signals, especially due to their responses to irregular natural events such as rainfall, flooding,
fire and extreme temperatures. While the traditional definition of the term *phenology* as the "*timing of*
*recurrent biological events*" (Lieth, 1974) is generally assumed to apply to seasonally cyclical events,
such as canopy bud burst and senescence in winter-deciduous forests, a closer look at life patterns in
many Australian biomes yields a more complex picture of non-seasonal, yet still periodic, events as the
major drivers of phenology (Specht and Brouwer, 1975;Pook et al., 1997;Ma et al., 2015).
In this synthesis, we examine the drivers of plant phenological cycles across Australia, illustrate the
potential measurement tools available, and suggest future avenues of required research. At the national
scale, we demonstrate how satellite observations can be used to characterize broad scale phenological
variability across the continent, and discuss the major drivers underlying these patterns. At the
ecosystem scale, we highlight the value of using digital image archives obtained from phenocams
installed at a selection of contrasting OzFlux sites (Australian and New Zealand flux tower network,
see Beringer et al. (2016)), for examining the phenology of canopies and individual plants in Australian
ecosystems. We discuss how greenness information obtained from phenocam imagery can be compared
with flux tower estimates of gross primary productivity (GPP) and why such relationships are stronger
from some Australian ecosystems than for others. Finally, we assess the feasibility and effectiveness of



phenocams for continuously observing Australia's diverse ecosystems.
**2 Drivers of phenology in Australia**
Australian ecosystems include a diverse range of biomes spanning from tropical savannas and
rainforests in the north, to arid and semi-arid shrublands and grasslands in the centre and finishing with
temperate evergreen forests and woodlands in the south. Temperature is an important phenological
driver in Australian ecosystems (Chambers et al., 2013), just as it is in temperate deciduous and boreal
ecosystems in the northern hemisphere. However, Australian ecosystem dynamics are arguably
influenced more strongly by the availability of water, because 70 % of its land mass is classified as arid
or semi-arid (Chambers et al., 2013). The response of LAI and vegetation cover to rainfall events, mean
annual rainfall and the corresponding soil water availability is well documented (Pook, 1985;Specht
and Specht, 1989;Ellis and Hatton, 2008;Donohue et al., 2009;2013;Andela et al., 2013;Duursma et al.,
2016). Similarly, vegetation density closely follows climate gradients in average rainfall (Ellis and
Hatton, 2008;Donohue et al., 2009;Hutley et al., 2011;Ma et al., 2013). Large-scale variability in
climatic conditions governs vegetation distribution across the continent and most Australian vegetation
has adapted in some form around the need to maintain a balance between water access and water loss.
Among the exceptions to this concept are temperate and tropical rainforest ecosystems found along the
east coast of the continent, where rainfall is almost continually in excess of water requirement. In
contrast, C4 grass phenology is driven by wet season rainfall across Australia's tropical savanna
ecosystems whilst the C3 eucalypt overstory remains relatively static throughout the year, supported by
access to deeper soil water reserves (Eamus et al., 2002;Hutley et al., 2011;Ma et al., 2013). Recurrent
dry spells and droughts are often associated with high mortality events in south western Australia (Evans
and Lyons, 2013;Matusick et al., 2013) and in both riparian (Davies et al., 2008;Gehrke et al., 2006)
and non-riparian trees (Semple et al., 2010) in southeast Australia. Over longer timescales, non-seasonal
shifts in temperature and rainfall, as well as second order effects on soil water availability, have been
attributed to forest declines (Evans et al., 2013) and mortality (Evans and Lyons, 2013).
Fire events are often triggered by drought and dry conditions, which are not traditionally regarded as
phenological drivers but nonetheless have a cyclical character in many Australian ecosystems (Maier
and Russell-Smith, 2012). Most eucalypt forests in Australia suffer periodic burning and have adapted
strategies to allow prompt regrowth after fire (epicormic growth) to obtain a rapid photosynthetic
advantage over other vegetation competing for light (Hodgkinson, 1998;Burrows, 2008). Drought can
also trigger insect attacks, which have been observed to result in rapid canopy defoliation and ecosystem
carbon loss in a eucalypt forest in southeast Australia (Keith et al., 2012;van Gorsel et al., 2013).
Seasonal grazing due to insects and other arboreal browsers is another periodic disturbance in Australian
ecosystems that can inhibit new leaf production and greatly modify the canopy, thus thwarting
phenology responses observable in these ecosystems (Lowman, 1985;Specht, 1985;Melzer et al., 2000).





Temporal variations in leaf colour, although not as pronounced as in temperate deciduous forests, do
occur in evergreen species that dominate Australian ecosystems. For example, eucalypt leaves can
contain large amounts of anthocyanin, causing a reddening of the leaf surface (Sharma and Crowden,
1974). While there is no general consensus on why anthocyanins are present in leaves (Gould et al.,
2000), their accumulation is associated with the expansion and subsequent predation of young foliage
(Close and Beadle, 2003), nutrient deficits that affect photosynthesis (Terashima and Evans,
1988;Sugiharto et al., 1990) or as a protection against solar radiation (Gould, 2004). Accumulation of
foliar anthocyanin has been observed in leaves of many Australian plant species (Nittler and Kenny,
1976;Hodges and Nozzolillo, 1996;Kumar and Sharma, 1999;Close et al., 2000;2001a;2001b), as well
as reddening of adult leaves due to a changing anthocyanin to chlorophyll ratio when exposed to
stressors such as cold or drought (Close et al., 2001a;Stone et al., 2001;Barry et al., 2009). In other
cases, exposure to stressful processes can result in a decrease in leaf chlorophyll content, accompanied
by or the precursor to, visible chlorosis (Coops et al., 2004). These leaf reddening events are visible and
often occur in a cyclical manner, therefore they can be considered as phenological in nature and may
be identified by phenocams.
**3 Continental scale phenology from satellite sensors**
To characterize the phenological diversity of Australian ecosystems at the continental scale, we used
satellite remote sensing observations to map land surface seasonal variability in measures of vegetation
greenness. For this, we used EVI as a phenology indicator, whereby we constructed an EVI seasonality
map for Australia using 8-day composites of spectral reflectance observations from MODIS
(MCD43C4.005 product) at 0.05 degree (*c*. 5 km) resolution for the period 2000-2012. We calculated
EVI (Huete et al., 2002) as Eq. (1):
$$EVI = G \frac{\rho_{NIR} - \rho_{red}}{\rho_{NIR} + L + C_1 \cdot \rho_{red} - C_2 \cdot \rho_{blue}} \tag{1}$$
where $\rho_{NIR}$, $\rho_{red}$, and $\rho_{blue}$ are atmospherically corrected spectral reflectance in near-infrared, red and
blue wavelength ranges, respectively; G is a gain factor (G = 2.5); $C_1$ and $C_2$ are aerosol resistance
coefficients ($C_1$ = 6, $C_2$ = 7.5) ; and L is a soil-adjustment factor (L = 1). Values of EVI below 0 and
above 1 were removed, as these pixels were either contaminated by cloud or open water.
The mean seasonal pattern for each pixel, based on variability in EVI from 2000-2012, was calculated
as the mean annual pattern in 8-day values over the 13 years. Subsequently, this seasonal pattern was
subtracted from the time-series for each pixel to yield seasonally adjusted anomalies. We then calculated
(i) the standard deviation in these seasonally adjusted anomalies (n=585), as well as (ii) the overall
mean EVI and (iii) the standard deviation of the mean seasonal pattern (n=45). Combining these three
measures allowed us to distinguish areas in Australia where the phenological signal was primarily 'non-





seasonally dynamic' (EVI varied but not in accordance with seasons), 'constantly high' (EVI remained
high year-round), or 'seasonally dynamic' (EVI showed regular seasonal variability).
Our continental phenological response map shows the tropical region of northern Australia (Fig. 1; A
& B) experiences a predictable seasonal phenology, as it receives reliable summer monsoon rainfall
(Cook and Heerdegen, 2001). In contrast, large areas of Australia are characterised by non-seasonal
variability in vegetation cover (Fig. 1, C). These areas fall largely within the arid interior, as well as in
the south-western and south-eastern subhumid regions, where rainfall variability has been particularly
strong during the observation period, including the worst multi-year drought on record (Van Dijk et al.,
2013;Broich et al., 2014).
To explore phenological variability in more detail around Australia, EVI is displayed by timeseries plots
for selected sites (Fig. 1). Tropical savanna ecosystems in northern Australia depict both seasonal
(monsoon driven) and non-seasonal changes (fire driven), represented by EVI for Howard Springs
(location A). In contrast, the tropical rainforest at Cape Tribulation (location B) shows a regular but low
level of seasonal variability where the maximum EVI is in the late dry season, and the minimum EVI
is in the wet season (Fig. 1). An exception to this regular cycle is the impact of cyclone Larry in early
2006. Cape Tribulation is located nearby and within the same tropical rainforest as the Cow Bay OzFlux
site (Fig. 1). Both sites show the same seasonal trend and these changes in EVI can be related to changes
in NEE that are observed at the OzFlux towers at both sites (Beringer et al., 2016).
Further inland, floodplain vegetation in the arid Channel Country (location C) responds primarily to
infrequent and irregular flood events produced in the large upstream catchment. For a flood event in
2010, the time-series shows an initial reduction in EVI (open water produces negative EVI (Huete et
al., 2002)) with a subsequent green flush. The Western Australian wheat belt (location D) shows a
winter peak in vegetation, corresponding with maximum leaf area before the cropping of winter wheat.
In contrast to location C, the year 2010 was the driest on record with poor harvests, recognisable as a
lower EVI signal for that year at location D. Sub-tropical evergreen forest on the central NSW coast
(location E) show a strong seasonal cycle that is most likely a phenological adaptation to summer-
dominant rainfall (Bowman, 2000). Highland vegetation in Tasmania (location F) shows a summer
green phenology that is more reminiscent of temperate forest ecosystems in the northern hemisphere.
These selected examples suggest that for most, but not all, of Australia's ecosystems it is not
temperature or radiation but water availability and extreme events that drive vegetation phenology.
Furthermore, in addition to the more or less predictable seasonal variations in precipitation, there is
very strong variability in rainfall between years due to the strong influence of ocean circulation modes
such as the El Niño Southern Oscillation, Indian Ocean Dipole, Southern Annular Mode and Pacific
Decadal Oscillation (Van Dijk et al., 2013;Broich et al., 2014;Cleverly et al., 2016).



**4 Ecosystem scale phenology from phenocams**

Digital repeat photography has already been shown to provide high temporal resolution phenology information throughout many vegetation biomes in the northern hemisphere (i.e. Richardson et al. (2007);Migliavacca et al. (2011);Sonnentag et al. (2012);Keenan et al. (2014a);Klosterman et al. (2014);Toomey et al. (2015)). Images collected by phenocams are typically stored in JPEG format with red, green, blue (RGB) digital numbers (DN) for each pixel in the image. Raw image format can also be used, which is discussed in detail by Richardson et al. (2013b) and Brown et al. (2016). Variability in scene illumination can affect the RGB brightness levels (Woebbecke et al., 1995;Richardson et al., 2007), which can be minimised by using a transformation of the RGB numeric values to chromatic coordinates (Gillespie et al., 1987;Woebbecke et al., 1995;Sonnentag et al., 2012). Of these indices, the green chromatic coordinate (GCC) has been identified as the most relevant to green vegetation phenology, given by Eq. (2):

$$GCC = G\,/(R + G + B) \tag{2}$$

where R, G and B are the per-pixel DN stored in the image, recorded as arbitrary units of intensity by the camera's charge coupled device. GCC is calculated for each pixel of the image, and then averaged over a user-defined region of interest (ROI). The red (RCC) and blue (BCC) chromatic coordinates were also calculated, in the same way as GCC in Eq. (2).

An advantage of being able to transform the phenocam images into these quantitative indices is that it allows for automated quality control of the data. This is important when a large network of automated measurements is being considered at a continental and global scale (Brown et al., 2016). Automated QA/QC routines have been regularly used for the large-scale processing of network data (Taylor and Loescher, 2013;Beringer et al., 2016). Furthermore, these automated routines can allow for the fast calculation of online summary metrics that can be used for diagnosing instrument problems and highlighting potential flaws in data collection (Smith et al., 2014). As a result, instrument downtime can be minimized and data collection can be optimized across a large network of sensors.

**4.1 Insights from existing phenocams**

To demonstrate the utility of phenocams for monitoring *in-situ* vegetation phenology in Australian ecosystems, we calculated GCC from image data sets from four contrasting ecosystems in Australia; Cow Bay (AU-Cow, Fig 4; site 1), Howard Springs (AU-How, Fig 4; site 2), Tumbarumba (AU-Tum, Fig 4; site 3) and Whroo (AU-Whr, Fig 4; site 6). We used the R software package (R Core Team, 2013) to extract and calculate chromatic coordinates (Eq. 2) from the phenocam images. Temporal coverage varied for each camera, so we analysed all images and calculated a daily average value of GCC, RCC and BCC.



### 4.1.1 Tropical rainforest

In the EVI phenology map, the Daintree tropical rainforest region (Fig. 1; B) displays constantly high EVI, as greenness varies little throughout the year. Overstory phenology at the evergreen Cow Bay rainforest OzFlux site (AU-Cow) showed GCC of the entire tree canopy was not seasonally dynamic either (Fig. 2). However, six individual tree crowns selected as ROIs and analysed individually revealed a more dynamic variability in GCC (Fig. 2), which fluctuated in line with leaf shedding and flushing events in some trees (Tree 1, Species *Wrightia laevis* and Tree 2, *Acmena graveolens*) or remained relatively constant in others (Tree 3, *Dysoxylum alliaceum* and Tree 4, *Cerbera floribunda*). Species-rich ecosystems can include a wide range of phenologies (Wright and van Schaik, 1994;Reich et al., 2004) that become indistinguishable when averaged over the entire ecosystem. In contrast to temperate deciduous and boreal ecosystems in the northern hemisphere, where seasonal dynamics of the canopy are largely temperature-controlled, the changes evident from Cow Bay suggest that the different species respond to a variety of cues, resulting in an apparent evergreen canopy despite significant individual phenological variability (Wu et al., 2016).

Tropical rainforests in nearby southeast Asia often show little or no clear seasonal dynamics in canopy cover and productivity (Kho et al., 2013), but are well known for synchronous mast fruiting with a return frequency of around 2-10 years (Visser et al., 2011). Less understood are similar 'masting' events in the forests of the wet tropics of north Queensland (M. Bradford, *pers. comm.*). These masting events occur with a return frequency of around 7 years and the trigger appears to be drier than normal conditions. Another characteristic of the Wet Tropics area is cyclone activity during the wet season, where the coastal forests are known colloquially as 'cyclone scrub'. The return frequency of low intensity cyclones is around 5 years (Australian system: Category 1-for a coastal crossing) while more intense cyclones have longer return frequencies (Turton and Stork, 2008). To date, no long-term phenology studies have been published for the rainforests of Far North Queensland. The combination of phenocams and weather stations on the Cow Bay (AU-Cow), Cape Tribulation (AU-Ctr) and Robson Creek (AU-Rob) OzFluz towers, all parts of the FNQ Rainforest SuperSite (Fig. 1), will allow the timing and the drivers of synchronous masting events and effects of cyclone activity to be studied in detail over the next few decades.

To highlight the utility of phenocams for identifying individual phenological variability, we looked at one of the trees (Tree 1, *Wrightia laevis*, the most variable tree in Fig. 2) in more detail (Fig. 3). When leaf shedding occurred, GCC rapidly decreased with an associated rapid increase in BCC. The analysis also showed a slower rate of leaf flushing compared to the swift leaf fall event. At the onset to the rapid decrease in GCC, RCC increased markedly (Fig. 3). This increase in redness may be due to the build-up of anthocyanin pigments that cause the red colouration of senescing leaves, also a distinct feature of deciduous forest canopies in autumn (Hoch et al., 2001;Lee et al., 2003;Gould, 2004). The RCC response shows that phenocams can capture leaf reddening at fine spatial and temporal resolution,



thereby opening up opportunities for understanding the causes and effects of such changes in more
detail.

### 4.1.2 Tropical savanna

Unlike the constantly high EVI of Australia's tropical rainforests, savannas display seasonally dynamic
EVI at the regional scale (Fig. 1; A). Extraction of GCC for understory images revealed a strong
seasonal phenological response at the Howard Springs site, evident by a sharp increase in GCC at the
onset of the wet season (Fig. 4). In contrast, similar comparison of these indices for the overstory images
showed that overstory dynamics varied much less than those of the understory (Fig. 4). There are two
main strategies employed by plants growing in seasonally dry savannas; drought avoidance through
deciduous or die-back phenology and drought tolerance through evergreen phenology (Williams et al.,
1997;Tomlinson et al., 2013). We observed both strategies, as annual grasses displayed a boom bust
cycle in GCC whereas the eucalypt dominated overstory maintained its leaf cover without major
variability in GCC. However, the savanna overstory at this site also includes a small portion (*c.* 20 %)
of semi-, brevi- and fully deciduous species that shed their leaves primarily in the dry season (Williams
et al., 1997;Hutley et al., 2011). The phenocams at this site did not capture these species, highlighting
the need to consider phenocam positioning and field of view (FOV) when installing in complex
ecosystems.
The strong seasonal changes seen in the understory (Fig. 4) are a characteristic phenological response
of savanna grasses to the onset of the dry season. Each year, the commencement of the dry season
triggers understory C4 grass senescence (Andrew and Mott, 1983) and canopy leaf fall (Williams et al.,
1997) as the plants prepare to survive through the rainless months of May to September (Cook and
Heerdegen, 2001). This phenological change results in a transition from green to brown in the
understory, as evidenced by reduced GCC and a reduction of savanna GPP (Whitley et al., 2011;Ma et
al., 2013;Moore et al., 2016). The reduction of leaf water content associated with senescence also causes
an increase in fire susceptibility. During the dry season, the senesced understory is consumed in fire
events at 1-5 year intervals (Beringer et al., 2015). Fire can affect phenology in the short term through
its impact on canopy area (Cernusak et al., 2006;Beringer et al., 2007) and over longer timescales
through feedbacks to plant demography (Beringer et al., 2011;Werner and Franklin, 2010;Werner and
Prior, 2013). In addition to fire, cyclone activity can disturb Australia's savannas with wind throw from
severe tropical storms resulting in patches of defoliation (roughly every 5 years) and extreme cyclones
causing up to complete destruction (once every 500-1000 years) (Hutley et al., 2013). In Australian
savannas, pronounced spatio-temporal variability exists in this phenology, which still requires further
examination in fine detail so it can be more accurately understood (Ma et al., 2013).



**4.1.3 Temperate evergreen forest**
We calculated GCC for the Tumbarumba OzFlux site (AU-Tum), a temperate evergreen forest that
represents another key Australian biome. Two phenocams were installed at this site, one on the flux
tower at 60 m with a nadir view angle and the other at 1.5 m with an oblique view angle. As a result,
the cameras measured different ROI's. Despite the differences in orientation and view, GCC values
from both phenocams at the Tumbarumba wet sclerophyll forest clearly increased from November 2014
to January 2015, showing a flush of the understory leading into summer (Fig. 5).
For wet sclerophyll ecosystems in Australia, like the forest at Tumbarumba, flushing events in summer
are a common occurrence (Restrepo-Coupe et al., 2015), as temperature and light tend to be the primary
drivers of phenology, with water availability a secondary factor (van Gorsel et al., 2013;Rawal et al.,
2014). Tumbarumba is situated in a subalpine zone where winter minimum temperatures often drop
below zero, with occasional snow events (van Gorsel et al., 2013). Low temperature and incoming solar
radiation in winter result in reduced vegetation cover and activity in alpine and subalpine regions,
whereas increased radiation and temperature in summer promotes rates of vegetation emergence,
flowering and cover (Law et al., 2000;Venn and Morgan, 2007;Green, 2010). A recent study from
Rawal et al. (2014), looking at environmental effects on eucalypt phenology, found that photoperiod
length and temperature were key variables controlling the growth rates of several eucalypt species in
wet and dry sclerophyll forests in southeast Australia. While this notion holds true for wet sclerophyll
forests such as Tumbarumba (Keith et al., 2012;van Gorsel et al., 2013), water availability increases in
importance as a phenology driver in dry sclerophyll forests (Rawal et al., 2014;Duursma et al., 2016).
Continued phenocam monitoring at Tumbarumba and other dry sclerophyll sites (i.e. Whroo &
Cumberland Plain, Fig. 1), as well as the addition of more cameras in wet and dry sclerophyll forests,
will improve our understanding of the phenological processes occurring at temperate evergreen sites
over longer timescales in Australia.
**4.2 Phenocams and ecosystem productivity**
Whilst phenocams are clearly useful for identifying fine scale phenological changes in terrestrial
ecosystems often missed by larger scale satellite indices, the extracted indices can also provide a useful
comparison with productivity estimates, such as GPP. Toomey et al. (2015) compared phenocam
derived GCC to eddy covariance derived estimates of GPP at several sites within North America and
Canada and found correlation between some ecosystem GCC indices and GPP. They concluded that
phenocams can be a valuable tool for independently verifying variability in GPP attributable to
phenology. To demonstrate the potential of this idea for Australian ecosystems, we plotted GPP data
from two contrasting Australian ecosystems against phenocam derived GCC. At the first site, Howard
Springs tropical savanna (GPP data from Moore et al. (2016)), GCC varied in line with that of GPP,
both of which reach a maximum at the peak of the wet season (Fig. 4). The GCC signal rapidly decreases
once the predominant understory grasses senesce at the end of the wet season, and is also reflected by



a reduction in GPP (Fig. 4). Whitley et al. (2011) attributed increased LAI in the understory to the rapid
increase in savanna GPP. Likewise, Moore et al. (2016) attribute the seasonal dynamics of savanna GPP
to be largely determined by understory flushing, with a more steady contribution from the overstory
throughout the year. Fig. 4 supports the conclusions of Whitley et al. (2011) and Moore et al. (2016) as
it shows a much more dynamic understory when compared with the overstory, the cycles of which are
reflected in the ecosystem GPP estimate.
At the second site, Whroo evergreen dry sclerophyll woodland (GPP data from Beringer et al. (2016)),
GCC and GPP  do not track each other over time like at Howard Springs (Fig. 6). While GCC remains
relatively constant, there is a clear growing season displayed by GPP in the summer months (i.e. Dec-
Feb, Fig. 6). In evergreen ecosystems such as Whroo, productivity is driven more by vegetation
responses to meteorological drivers (i.e. solar radiation, air temperature, rainfall) rather than to
phenological variability (Restrepo-Coupe et al., 2015). Leaf area at Whroo is maintained at
approximately 1.0 m$^{-2}$ m$^{-2}$ throughout the year (I. McHugh, *pers. comm.*) and being evergreen in nature,
the phenocam is not able to discern phenological variability related to greenness at the ecosystem scale.
For temperate evergreen ecosystems in Australia, phenocams are not as useful for linking with GPP
(Restrepo-Coupe et al., 2015), but still likely hold value for identifying individual phenology signals
within the evergreen canopy.
**5 Expanding the Australian Phenocam network**
We have shown the value of phenocams for ecosystem monitoring in Australia, how they can be used
to inform on species level changes, supplement large scale satellite remote sensing data and aid in
interpreting ecosystem GPP. In Australia, phenocams have primarily been installed at several pre-
existing    Terrestrial    Ecosystem    Research    Network    (TERN)    supported    OzFlux    sites
(http://www.ozflux.org.au/), at which eddy covariance flux towers are used to study carbon, water and
energy exchanges between ecosystems and the atmosphere. Phenocams are currently being deployed
across TERN and the Australian SuperSite Network (http://www.tern-supersites.net.au), where a
comprehensive set of co-located measurements of vegetation, faunal biodiversity, soil/water and remote
sensing are being made. Despite these recent deployments, the spatial and temporal coverage of
phenocams in Australia remains limited (Fig. 1).
Several of the phenocams in Australian ecosystems are either in the early stages of operation or have
experienced issues with long term stability in data collection (i.e. field-of-view shifts, lack of timestamp
standards; loss of data) and some ecosystems in Australia are missing altogether (i.e. alpine ecosystems,
managed agricultural land, mangroves, Fig. 1). Phenocams are one of few observing methods that can
bridge across spatial and temporal scales, from individual plants to the ecosystem and continental scales,
and between biophysical function and ecological condition and composition. They can capture seasonal
trends (Fig. 4, 5 & 6) and short lived events (Fig. 2 & 3) that are often missed by coarse scale techniques



(i.e. satellite sensors, Fig. 1). As such, establishment of a national phenocam network in Australia will
be an extremely valuable contribution to help integrate data products between TERN Facilities
addressing different ecosystem questions at different scales (i.e. the SuperSites, OzFlux, LTERN,
AusCover and eMAST).
**5.1 Data standardisation and sharing**
Standardization and quality control in phenocam data collection is essential to support cross-site and
cross-ecosystem comparisons. There is a clear need for standards in measurement in Australia, with a
wide range of phenocam hardware currently in use. Brown et al. (2016) provide recommendations for
improving phenocam datasets and improving comparability between sites, including:
•   maintaining metadata and data management standards (i.e. image naming, FOV, camera

11         settings) for all camera-based data

•   registering all publicly available phenocams with a regional phenocam network
•   making datasets available online wherever possible
Analysis of some of the datasets revealed that camera quality and installation issues (i.e. sun glare,
inappropriate image acquisition times, power consumption) play a role in data quality. However, the
single greatest challenge was found to be changes in data quality and characteristics over time that limit
or prevent automated analysis. In particular, changes in the FOV over time due to intentional or
unintentional reorientation of the camera, creates a major obstacle to the collection of a sufficiently long
time series and to automated data analysis. Slight or gradual changes can be dealt with through
additional image co-registration efforts, but they can be arduous and reduce the area effectively
available for analysis. Solid mounting of the camera and, when necessary, accurate re-alignment after
removal (i.e. for maintenance) or unintentional reorientation can help alleviate such issues. Maintaining
a consistent FOV is critical. For all but the most homogeneous environments, the FOV of the phenocam
*is* the dataset. Each time the camera moves, the usable pixels for which there is long term monitoring
data become increasingly constrained.
Data publication and sharing are also important factors to consider. We have formed the Australian
Phenocam Network (http://phenocam.org.au/) to provide a platform for phenocam data sharing, storage
and publication to the wider scientific community. This network is continually evolving, with data made
available under the TERN 'By Attribution' license based on the Creative Commons framework
(http://www.tern-supersites.net.au). As for other national and global multi-site data sets, open access to
phenocam data will enhance the reproducibility and extensibility of research and the combined value
of such data (Brown et al., 2016).



## 6 Conclusion

We see considerable potential for the developing Australian phenocam network. Existing infrastructure, supported by TERN, will assist in the establishment of this network, the implementation of more phenocams around the country and the standardisation of data collection, analysis and sharing. An Australian phenocam network would also provide valuable data for monitoring networks such as OzFlux, SuperSites, AusPlots and eMAST. Currently, there are only a small number of continuously operating phenocams in Australia, with even fewer uploading data to the Australian phenocam website. Expansion of the currently sparse distribution of Australian phenocams will help improve our understanding of the diversity of phenological strategies employed by Australian ecosystems. Combined with satellite remote sensing techniques, phenocams can improve our ability to quantify and predict the functioning of Australia's ecosystem within the Earth system.

## Acknowledgements

First and foremost, the authors would like to acknowledge the Australian Centre for Ecological Analysis and Synthesis (ACEAS) for their support of a workshop aimed at developing a Phenocam Network in Australia, held on Stradbroke Island in March 2014. Support for collection and archiving of some of the data used in this paper was provided through the Australian Government Terrestrial Ecosystem Research Network (TERN) (http://www.tern.org.au) Facilities: ACEAS, Auscover, SuperSites, OzFlux and eMAST.  Part of the data was also funded via Australian Research Council grants DP0772981 and DP130101566. Beringer is funded under an ARC Future Fellowship (FT1110602). Keenan acknowledges support from a Macquarie University Research Fellowship. TBR cameras were founded by a Re-establishment Grant 2013 from the University of Technology Sydney: "Setup of a phenocam network on some key Australian ecosystems" (N. Restrepo-Coupe, CI). Moore also personally thanks Matthew Northwood, from Charles Darwin University, for his assistance with the Howard Springs phenocams.

## Authorship Note

This paper was conceived and outlined at the ACEAS 2014 Phenocam Workshop led by Brown and Keenan. Moore created the first draft and was the primary editor for subsequent drafts. Keenan and Brown provided oversight and additional edits. All authors provided additional text, edits and review. Liddell provided data for the Daintree analysis, van Dijk provided analysis of the remote sensing data, and Duursma provided primary analysis for the Daintree and Tumbarumba data. Moore provided data and analysis for the Howard Springs data and Culvenor and provided data for Tumbarumba. Restrepo-Coupe and provided data and analysis for Daintree, Whroo and Tumbarumba dataset.

Figure Credits: 1: van Dijk & Moore; 2&3: Duursma; 4: Moore; 5: Culvenor & Moore; 6: Moore & Restrepo-Coupe




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



**Figure 1:** Enhanced Vegetation Index (EVI) map of Australia obtained from MODIS global 0.05 degree 8-day reflectance composites for 2000-2012 (MOD09) with a selection of six pixels to demonstrate the following: Yellow areas have a strong seasonal cycle, such as Howard Springs at point A. Cyan areas have strong, but non-seasonal variations in EVI, such as Channel country at point C. Magenta have constant moderate to high EVI with relatively little temporal variability, such as Cape Tribulation at point B and the central coast at point E. Green areas have both seasonal and non-seasonal variation, such as the wheat belt of WA at point D. Orange areas have relatively high greenness but also a seasonal component, such as the Tasmanian highlands at point F. Blue areas have relatively high





greenness but strong non-seasonal variation. The paler colours of the map centre show weaker EVI signals. The SD in seasonally adjusted anomalies (yellow) was scaled from 0-0.1; SD in mean seasonal cycle (cyan) from 0-0.1; and mean EVI (magenta) from 0-0.6. Also included are the locations (and Fluxnet codes where applicable) of currently operating phenocams in Australia. Numbers refer to; 1. Cow Bay (AU-Cow), 2. Howard Springs (AU-How), 3. Tumbarumba (AU-Tum), 4. Karawatha, 5. Great Western Woodlands (AU-GWW), 6. Whroo (AU-Whr), 7. Wombat (AU-Wom), 8. Robson Creek (AU-Rob), 9. Litchfield (AU-Lit), 10. Warra (AU-Wrr), 11. Calperum (AU-Cpr), 12. Cumberland Plain (AU-Cum) & EucFACE, 13. Alice Springs (AU-ASM), 14. Riggs Creek (AU-Rig), 15. Sturt Plains (AU-Stp), 16. Cape Tribulation (AU-Ctr).



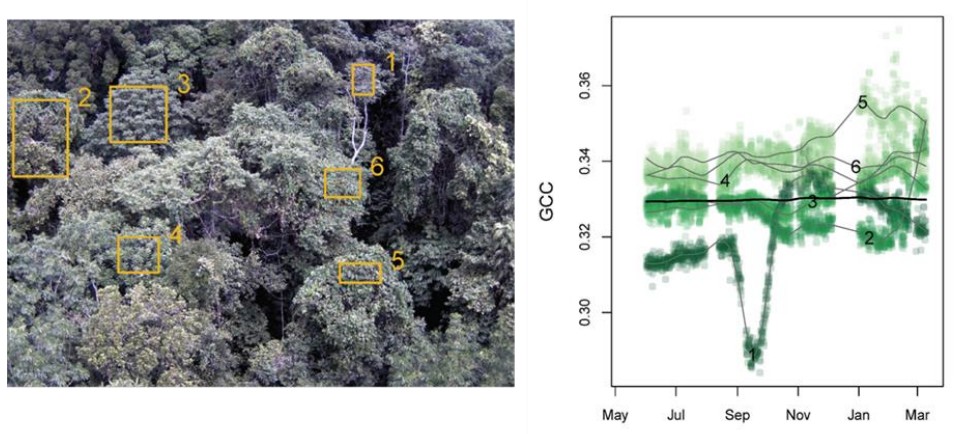

**Figure 2**: Green chromatic coordinates (GCC) for six individual tree crowns (indicated by orange regions of interest) at the Cow Bay tropical forest OzFlux site, Queensland, Australia. Leaf shedding and flushing events (note especially Tree 1, *Wrightia laevis* and Tree 2, *Acmena graveolens*) are highly specific to individual trees. Black line represents GCC from the full camera field-of-view.




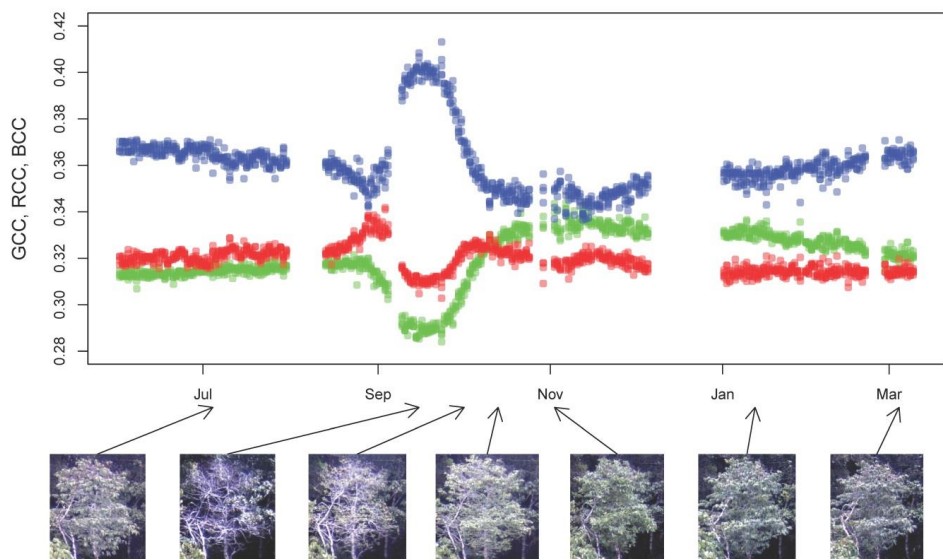

**Figure 3:** Green (GCC), red (RCC) and blue (BCC) chromatic coordinates for a single tree (*Wrightia laevis*) at the Cow Bay tropical forest OzFlux site, Queensland, Australia. Quick leaf shedding and flushing events are well captured by the phenocam.



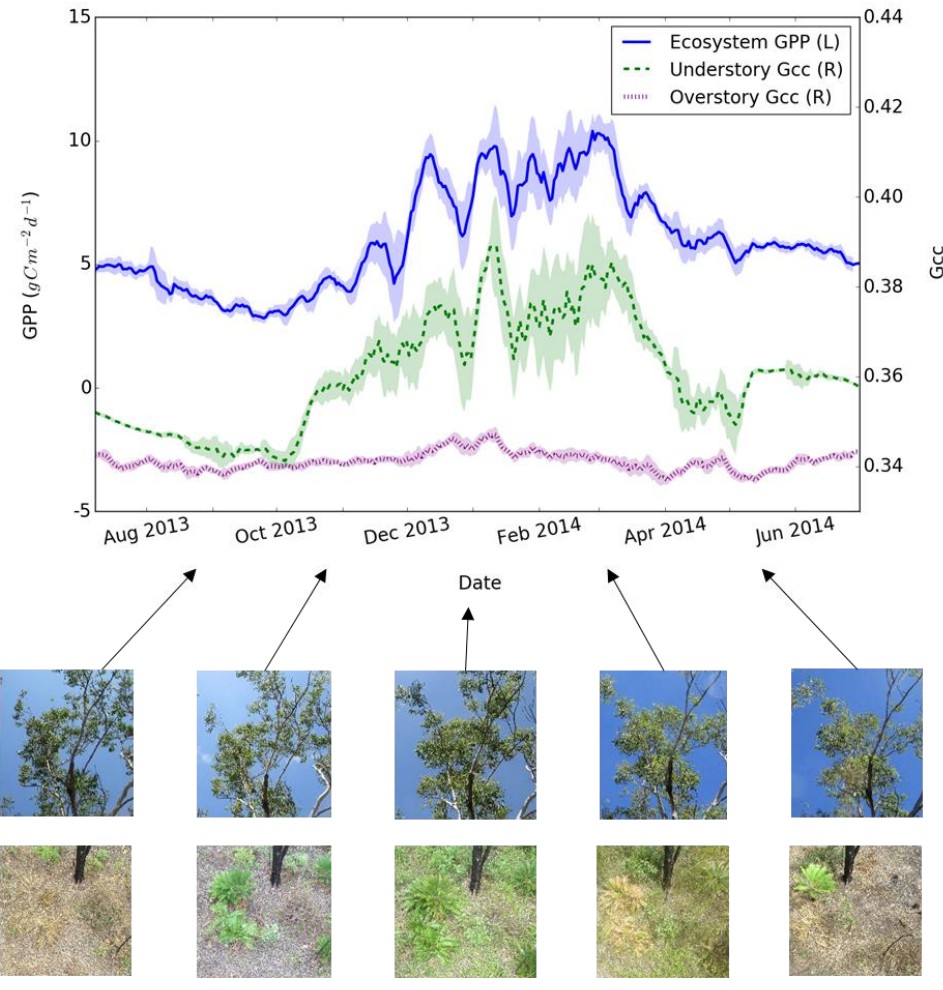

**Figure 4:** Savanna ecosystem gross primary productivity (GPP) and green chromatic coordinates (GCC) for the overstory and understory, with region of interest (ROI) image examples, at Howard Springs OzFlux site, Northern Territory, Australia. Data are shown as an 8-day centred running mean with ± 90 % confidence shading.





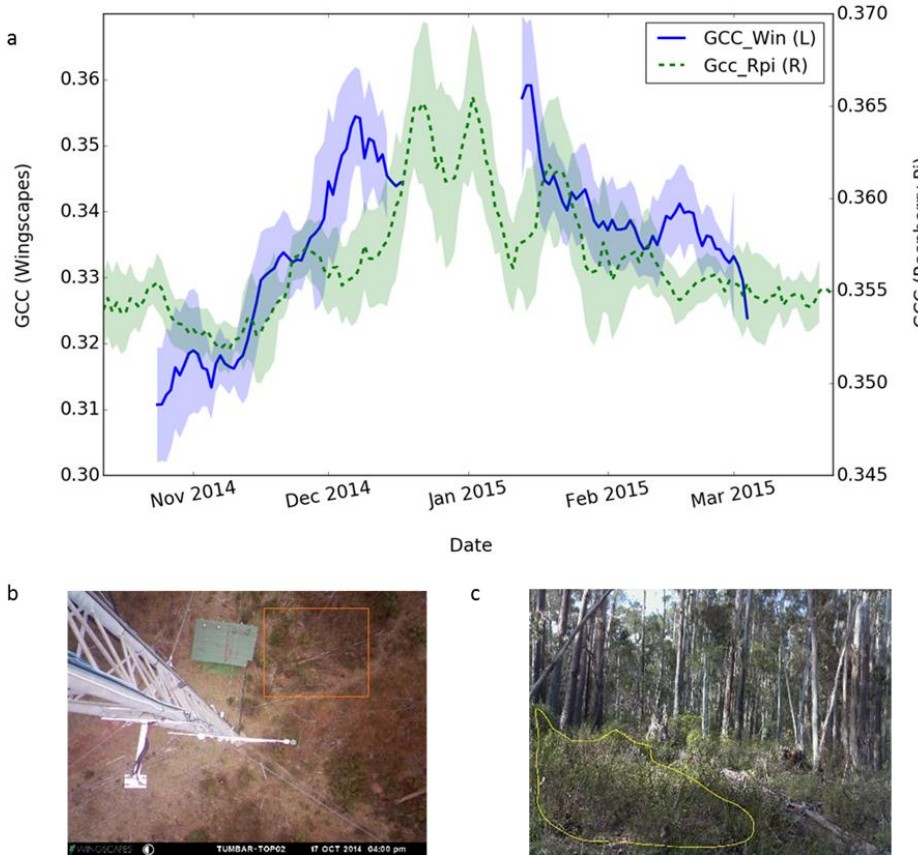

**Figure 5:** Phenocams in use at Tumbarumba (AU-Tum), New South Wales, Australia showing a) green chromatic coordinates (GCC) for understory vegetation from b) a downward facing wingscapes phenocam (region of interest inset) and an oblique facing raspberry pi (Rpi) phenocam (region of interest inset). GCC data are shown by an 8-day centred running mean with ± 90 % confidence shading.



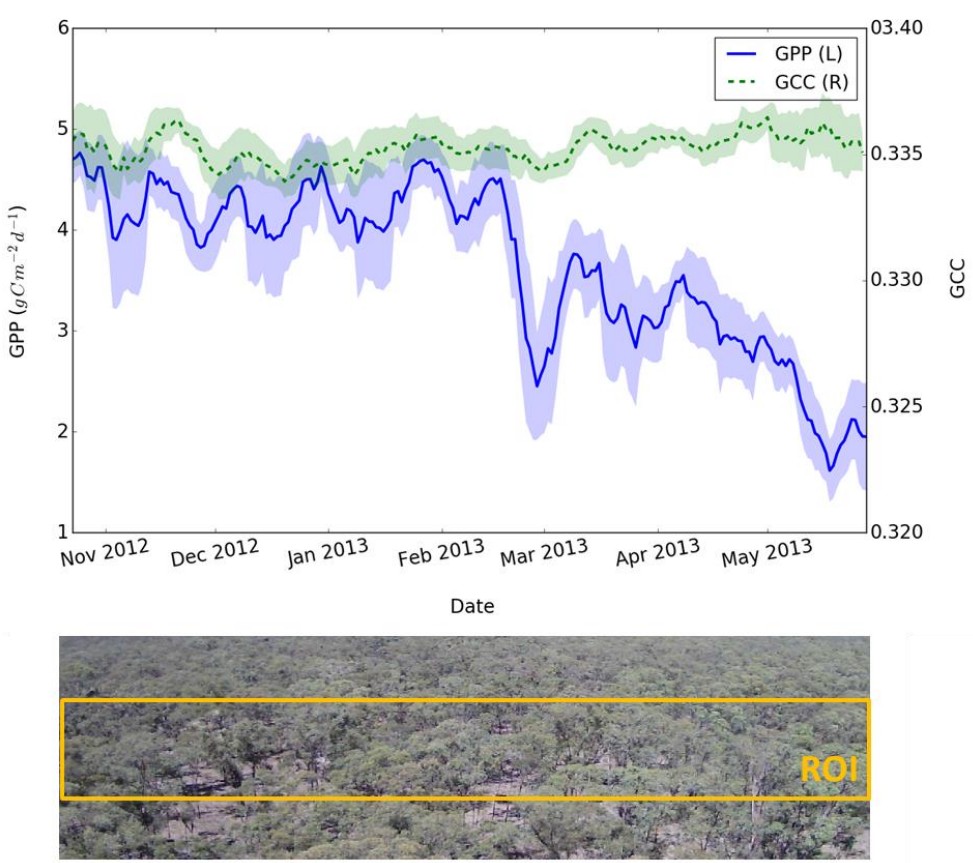

**Figure 6:** Ecosystem gross primary productivity (GPP) and green chromatic coordinate (GCC) index for a temperate eucalypt woodland (Whroo Conservation Reserve) in southeast Victoria, Australia. Both GPP and GCC data are shown by an 8-day centred running mean with ± 90 % confidence shading. The region of interest (ROI) for GCC calculation is indicated by the orange box in the image.