# Peer review of "Australian vegetation phenology: new insights from satellite remote sensing and digital repeat photography"

_Biogeosciences, 2016_

## Referee Comment (RC1) · Anonymous Referee #1 · 12 May 2016

Dear authors,

Authors well surmmarised the plant phenology in Australia by analysing the satelliteand in situ-observed remote-sensing data. They also focused on the PhenoCam activities in Australia. I was exited to read and inspired many ideas. From the plant climatological view point, the long-term continuous phenological observations by using time-lapse cameras and satellite remote-sensing are pretty important. I expect further studies such as the evaluation the relationship between spatio-temporal variability of plant phenology and climate change in Australia. Please conduct further studies based on the current study in near future. The manuscript was written well. So, I'd like to recommend to accept this manuscript after the following minor revisions.

1: I think it's not so easy for readers who have no background of climatology and ecology in Australia. Please show the map of annual mean air temperature, annual precipitation, Coppen climate classification, and land cover classification.

2: page 7, line 25: Please explain "NSW".

3: page 8, line 21: Please explain "QA/QC".

4: page 8, line 32: Please explain "RCC and BCC".

5: page 9, line 15: In Malaysia, the general flowering was occurred after sever dryness events. Please see the following paper.

Sakai, S., Harrison, R.D., Momose, K., Kuraji, K., Nagamasu, H., Yasunari, T., Chong, L., Nakashizuka, T., 2006. Irregular droughts trigger mass flowering in aseasonal tropical forests in Asia. American J. Botany 93(8), 1134–1139.

In addition, the following paper analysed Gcc in a tropical rainforest in Malaysia.

Nagai S, Ichie T, Yoneyama A, Kobayashi H, Inoue T, Ishii R, Suzuki R, Itioka T (2016) Usability of time-lapse digital camera images to detect characteristics of tree phenology in a tropical rainforest. Ecological Informatics, 32:91–106.

6: Page 9, lines 29-36 (Fig. 2): Can you explain the reason of characteristics of tree phenology in each species?

7: page 15, kine 7: Garbling? 1988–2008

8: page 24: Please explain "WA".

9: Figs 2-6: Please show the Coppen climate classification, the location of each site on Fig. 1, the site ID shown in Fig. 1 (number).

10: Figs 2,5, and 6: Please show the typical phenology images throughout a year.

11: Figs 4-6: Please explain "(L)" and "(R)".
12: Fig 5: wingscapes; Camera name?

13: Fig 5: GCC\_Win, Gcc\_Rpi -> Understory Gcc, Overstory Gcc Best wishes,

---

## Referee Comment (RC2) · Anonymous Referee #2 · 27 May 2016

**GENERAL COMMENT**

1. The manuscript presents a very interesting overview on the phenology of Australian ecosystems tracked from MODIS EVI and phenocam GCC. Potential and opportunities related to overstory and understory GCC analysis as well as comparison between GCC and ecosystem productivity are illustrated and discussed. Moreover the growing and promising Australian phenocam network is presented together with important recommendations on data standardization and sharing. The topic is relevant, timely and can be of great interest to the growing community of sites and researchers using phenocam for phenological and ecological monitoring

I only have one criticism related to EVI trajectories discussion in section 3, 4 and figure

1. EVI trajectories in fig1B and 1E does not look like "constant moderate to high EVI with relatively little temporal variability" (fig1 caption). Or at least I would not define 1B and 1E with little temporary variability and 1F with "a seasonal component" (fig1 caption). Qualitatively evaluating annual cycle amplitude looking at fig1, it can not be stated that site F is different from E and B. Mean EVI values are different between the 3 sites but mean annual amplitudes are quite similar. I think that saying that B and E does not show a season cycle is not correct. Moreover fig1 caption (little temporal variability at point B Cape Tribulation and point E) and p7 I13-14 are in contrast with section 3 p7 I3-4 and p7 I26-27 "(location E) show a strong seasonal cycle". A more quantitative approach to define what is high, low and null seasonal variability is needed. This can be quite easily done computing mean annual EVI amplitude. Section 3, section 4.1.1 (p9 I3) 4.1.2 (p10 I4) need to be modified accordingly.

I recommend manuscript publication after the above mentioned point and the following specific comments are addressed.

**SPECIFIC COMMENTS**

p7 I10-11 this sentence should go before the previous paragraph, where fig1 is mentioned.

p7 I14-15 within year patterns (e.g wet season) are difficult to see in the current plot. From the lower panels it's almost only possible to see inter-annual patterns rather than events occurring in specific period of the year e.g. "maximum EVI in the late dry season". Even if not extremely appealing, a vertical dashed grid at x axis ticks could help

p7 I23 I admit that I could be biased, but maybe adding months in parentheses, (e.g. winter (jun-sep)), should help readers from the northern hemisphere.

p8 l4-5 reference formatting issues

p8 I18-25 phenocam QA/QC is a relevant topic that is worth to be raised, but this paragraph is a bit misleading as mentioned references are not related to phenocam
QA/QC that to my knowledge are still missing. Please reformulate.

p8 I30 which R package? Or simply R software?

p8 I29-30 check figure numbering

p8 I32-33 Phenocam data normally need to be filtered using approaches a bit more sophisticated than daily averaging. Please comment on this in the light of commonly used filtering procedures (e.g. Sonnentag et al. 2012, Filippa et al. 2016).

p9 I6-7 & I31 how can you say that "GCC fluctuated in line with leaf shedding and flushing". I guess shedding and flushing were evaluated by visually inspecting the images. If yes you should mention it.

p10 I7 & I19 insert the month when the onset of the wet season and onset of the dry season occur. Probably Oct-Nov and Mar-Apr?

p10 I5-I17 MODIS EVI and understory GCC show pronounced seasonal cycles, whilst overstory GCC did not. Which is the overstory fractional cover? Can low fractional cover be the reason to explain why MODIS EVI matches undestory phenology rather than overstory?

p10 I24-I33 Are those longer term phenological patterns (fire and cyclone activity) detectable from EVI timeseries?

p11 I2-I24 In these paragraph it seems like temperate evergreen forest, wet sclerophyll ecosystem and eucalypt forest are used as synonyms. Is this correct? Try to be more consistent or make a short introduction in the paragraph to help readers not familiar with Australian ecosystems.

p11 l22 fig1 E?

p11 I1-I7 and fig5. Greening ramps of the two ROIs from late nov to late dec, show approximately a 1 month lag. Could this be related to understory phenological variability? Are the two ROIs looking at the same individuals?
p12 I7-8 here you refer to the site as an "evergreen dry sclerophyll woodland" while in fig 6 caption "temperate eucalypt woodland" is reported. Is it the same? Be more consistent.

Fig1 and caption. Consider the idea of plotting phenocam site whose date are used in paper (e.g. AU-How, presented in the paper differently form phenocam site not used.

Fig2 Does different green intensity has a meaning?

Fig4&6 legend. What does L and R means? It indicates left and right y axes? If yes it's not needed.

Fig4 pics in the lower panel: are these the ROIs used to compute overstory and understory GCC?

Fig3-5 including ecosystem type in figure caption or plot titles consistent with 4.1 paragraph titles (tropical rainforest, tropical savana, temperate evergreen) will help the reader.

**Cited references**

Sonnentag, O., Hufkens, K., Teshera-Sterne, C., Young, A. M., Friedl, M., Braswell, B. H., ... Richardson, A. D. (2012). Digital repeat photography for phenological research in forest ecosystems. Agricultural and Forest Meteorology, 152, 159–177. http://doi.org/10.1016/j.agrformet.2011.09.009

Filippa, G., Cremonese, E., Migliavacca, M., Galvagno, M., Forkel, M., Wingate, L., ... Richardson, A. D. (2016). Phenopix: A R package for image-based vegetation phenology. Agricultural and Forest Meteorology, 220, 141–150. http://doi.org/10.1016/j.agrformet.2016.01.006

---

## Author Comment (AC1) · 25 Jun 2016

Author response to reviewer comments for "Australian vegetation phenology: new insights from satellite remote sensing and digital repeat photography"

We thank both reviewers for their comments and suggestions about our manuscript and provide the following as our interactive responses to their points.

**Reviewer 1**

1: I think it's not so easy for readers who have no background of climatology and ecology in Australia. Please show the map of annual mean air temperature, annual precipitation, Coppen climate classification, and land cover classification.

To address this comment, we can generate a figure that includes each of the mentioned parameters for Australia. This figure can be addressed and discussed in section 1 (introduction) and section 2 (drivers of phenology in Australia) in the manuscript, to better contextualise Australian climatology and ecology. However, this manuscript is submitted as part of the OzFlux special issue in Biogeosciences. The manuscript: An introduction to the Australian and New Zealand flux tower network – OzFlux by Beringer et al. does provide a biome classification map of Australia. We could also refer to this overview paper to addressing the above comment.

2: page 7, line 25: Please explain "NSW".3: page 8, line 21: Please explain "QA/QC".4: page 8, line 32: Please explain "RCC and BCC". 8: page 24: Please explain "WA".

Each of these points refers to acronyms we failed to fully explain. Therefore, we will amend these in the manuscript. For the reviewer's reference, NWS refers to New South Wales, a state within Australia. Likewise, WA refers to another state, Western Australia. QA/QC means quality assurance and quality checks. RCC and BCC are similar to GCC in that they are the red chromatic coordinate and blue chromatic coordinate, respectively.

5: page 9, line 15: In Malaysia, the general flowering was occurred after sever dryness events. Please see the following paper.

Sakai, S., Harrison, R.D., Momose, K., Kuraji, K., Nagamasu, H., Yasunari, T., Chong,L., Nakashizuka, T., 2006. Irregular droughts trigger mass flowering in aseasonal tropical forests in Asia. American J. Botany 93(8), 1134–1139.

In addition, the following paper analysed Gcc in a tropical rainforest in Malaysia. Nagai S, Ichie T, Yoneyama A, Kobayashi H, Inoue T, Ishii R, Suzuki R, Itioka T (2016) Usability of time-lapse digital camera images to detect characteristics of tree phenology in a tropical rainforest. Ecological Informatics, 32:91–106.

To address this comment, we will add the following to our discussion on page 9:
Tropical rainforests in nearby wet equatorial Asian rainforest (Malaysia, Indonesia) often show ambiguous seasonal patterns in canopy cover and productivity (Kho et al., 2013), but are well known for synchronous mast fruiting with a return frequency of around 2-10 years (Visser et al., 2011). The general flowering in these forests that is associated with these masting events has been shown to be triggered by irregular droughts (Sakai et al., 2006). More recently phenocams have been used to analyse the phenology of a dipterocarp canopy, a forest type associated with mast events, in Borneo (Nagai et al., 2016). This study confirmed that indices such as %RGB and green excess index (GEI) can be used to track flowering and leaf flushing at the individual tree level. Less understood are similar 'masting' events in the forests of the wet tropics of north Queensland (M. Bradford, pers. comm.).

6: Page 9, lines 29-36 (Fig. 2): Can you explain the reason of characteristics of tree phenology in each species?

The species discussed in this section is Wrightia laevis (likely ssp. millgar). The leaf phenology of the local species is not described in the literature but the genus has deciduous characteristics which are reported in related Indian species. The local species is found in the following vegetation types: Semi-deciduous and deciduous notophyll vine forest. (BVG1M: 2d) 3.8.5b Torres Straight http://www.ehp.qld.gov.au/ecosystems/biodiversity/regionalecosystems/details.php?reid=3.8.5 Type 35 Tall semi-deciduous notophyll vine forest of structured red and yellow earths. Metamorphic hillslopes, southern Cape York Peninsula http://www.rainforest-crc.jcu.edu.au/publications/rainforests\_capeyork\_4.pdf

These vegetation types are found in an area (Cape York and Torres Straight) which has a more pronounced monsoon seasonality and it would seem likely that deciduous character is an adaptive advantage for these plant communities. At Cow Bay in the Daintree Wrightia laevis is not a common species, there are 4 individuals in the 1Ha census plot where the phenocam tower is located. A useful recent reference on seasonality in leaf phenology in tropical rainforest species may be found in Wu et al. 2016. Leaf deBGD
velopment and demography explain photosynthetic seasonality in Amazon evergreen forests http://science.sciencemag.org/content/351/6276/972. Therefore, we will add the following to page 9: L30: ..in more detail (Fig. 3). This species, Wrightia laevis, may be found further north in semi-deciduous and deciduous vegetation communities that are connected floristically to the rainforests of the Daintree region.

7: page 15, kine 7: Garbling? 1988–2008

This must be due to a glitch in the citation used in our reference manager, which we will fix for resubmission.

9: Figs 2-6: Please show the Coppen climate classification, the location of each site on Fig. 1, the site ID shown in Fig. 1 (number).

This is a great suggestion and we will add this information to each of figures 2-6 in the resubmission.

10: Figs 2,5, and 6: Please show the typical phenology images throughout a year.

We did not include the phenology images throughout the year in the mentioned figures as we thought the example images of the regions of interest represented typical images collected. However, we will adjust the figures to include more temporal phenocam images accordingly.

11: Figs 4-6: Please explain "(L)" and "(R)".

The L and R letters refer to the left y-axis and right y-axis, which we will clarify in the figure caption.

12: Fig 5: wingscapes; Camera name?

Yes, wingscape is the camera used, which we will clarify in the figure caption.

13: Fig 5: GCC\_Win, Gcc\_Rpi -> Understory Gcc, Overstory Gcc

We are not entirely sure what reviewer 1 means by this comment, as both GCC\_Win
and GCC\_Rpi refer to two different phenocams that recorded understory vegetation cover change at the Tumbarumba site, which is explained in the figure caption. However, we will review this figure caption and make sure it is clear that both cameras are recording understory vegetation cover change, not that of the overstory.

**Reviewer 2**

1. EVI trajectories in fig1B and 1E does not look like "constant moderate to high EVI with relatively little temporal variability" (fig1 caption). Or at least I would not define 1B and 1E with little temporary variability and 1F with "a seasonal component" (fig1 caption). Qualitatively evaluating annual cycle amplitude looking at fig1, it can not be stated that site F is different from E and B. Mean EVI values are different between the 3 sites but mean annual amplitudes are quite similar. I think that saying that B and E does not show a season cycle is not correct. Moreover fig1 caption (little temporal variability at point B Cape Tribulation and point E) and p7 I13-14 are in contrast with section 3 p7 I3-4 and p7 I26-27 "(location E) show a strong seasonal cycle". A more quantitative approach to define what is high, low and null seasonal variability is needed. This can be quite easily done computing mean annual EVI amplitude. Section 3, section 4.1.1(p9 I3) 4.1.2 (p10 I4) need to be modified accordingly.

Agreed. The reviewer is correct that the seasonal amplitude at sites B and E is not much less than that at some of the other sites. Better would be to describe their main distinction as having an EVI that is relatively high throughout the seasons, and we will change the caption and text accordingly.

1. p7 I10-11 this sentence should go before the previous paragraph, where fig1 is mentioned.

We will fix this in the resubmission.

2. p7 I14-15 within year patterns (e.g wet season) are difficult to see in the current plot. From the lower panels it's almost only possible to see inter-annual patterns rather
than events occurring in specific period of the year e.g. "maximum EVI in the late dry season". Even if not extremely appealing, a vertical dashed grid at x axis ticks could help

We can add a dashed vertical grid at the x-ticks to assist with this problem and amend the sentence highlighted. The EVI map is meant to show the general phenology pattern for the continent of Australia, and the time series plots for sites a - e demonstrate interannual variability. We will make this clearer in the text.

3. p7 I23 I admit that I could be biased, but maybe adding months in parentheses, (e.g. winter (jun-sep)), should help readers from the northern hemisphere.

We will address the references to seasons throughout the manuscript with the months in parentheses. The additional figure suggested by reviewer 1 should also assist in putting the Australian climate and ecology into perspective for international readers.

4. p8 l4-5 reference formatting issues

These must have been missed in our review before submission so we will fix these where applicable before resubmission.

5. p8 I18-25 phenocam QA/QC is a relevant topic that is worth to be raised, but this paragraph is a bit misleading as mentioned references are not related to phenocam QA/QC that to my knowledge are still missing. Please reformulate.

The reviewer raises a valid point here and we will rework the sentence in question to be more appropriate. The references used are in reference to other large data efforts that worked on QA/QC. We can clarify this and expand on this in the text.

6. p8 I30 which R package? Or simply R software?

We simply used R software, so will amend this line.

7. p8 l29-30 check figure numbering

BGD
This is a good pick up, the figure referenced should be Figure 1, not 4. We will fix this for the resubmission.

8. p8 I32-33 Phenocam data normally need to be filtered using approaches a bit more sophisticated than daily averaging. Please comment on this in the light of commonly used filtering procedures (e.g. Sonnentag et al. 2012, Filippa et al. 2016).

The reviewer raises an interesting point here. However, it is our opinion that there is yet no right or wrong way to filter phenocam data. Filippa et al 2016 tested numerous options for filtering and massaging phenocam data and found some worked better than others on different datasets for different reasons. The message of Sonnentag et al. 2012 was that daily averaging removed a lot of the variation in hourly GCC and gave decent values that required no further filtering. We felt that simple daily averaging (with a smoother applied for visualisation) allowed us to present our message for the different camera types used in different ecosystems across Australia.

9. p9 I6-7 & I31 how can you say that "GCC fluctuated in line with leaf shedding and flushing". I guess shedding and flushing were evaluated by visually inspecting the images. If yes you should mention it.

Yes, leaf shedding and flushing were determined visually. The phenocam images included in figure 3 does support this, so we will make more explicit reference to these images in the above mentioned sentence.

10. p10 I7 & I19 insert the month when the onset of the wet season and onset of the dry season occur. Probably Oct-Nov and Mar-Apr?

These dates approximate the wet and dry season, which will be included in the suggested location when we resubmit.

11. p10 I5-I17 MODIS EVI and understory GCC show pronounced seasonal cycles, whilst overstory GCC did not. Which is the overstory fractional cover? Can low fractional cover be the reason to explain why MODIS EVI matches undestory phenology
rather than overstory?

This is due to the highly dynamic nature of the understory grasses in the savanna ecosystem measured. These grasses are very productive in the wet season and then senesce in the dry season, which results in large seasonal variability in greenness. The overstory, on the other hand, is evergreen, so there is less variability in vegetation greenness. The overstory has a cover fraction of approximately 50 %, so EVI still followed the understory seasonality even at this moderate level of overstory fractional cover. We will further clarify this in the discussion of pg. 10, L5-17.

12. p10 I24-I33 Are those longer term phenological patterns (fire and cyclone activity) detectable from EVI timeseries?

Although this is an excellent and intriguing question, it is outside the scope of this shorter-term in situ based study. A longer study is needed using MODIS EVI, which we have not analysed here. However, we believe the coupling of phenocam imagery alongside satellite phenology products is a perfect example of why the two should be utilised in tandem. We will make this point clearer in the discussion on pg 10.

13. p11 I2-I24 In these paragraph it seems like temperate evergreen forest, wet sclerophyll ecosystem and eucalypt forest are used as synonyms. Is this correct? Try to be more consistent or make a short introduction in the paragraph to help readers not familiar with Australian ecosystems.

We will make sure the manuscript is clear and concise about what type of eucalypt forest we are referring to. Two different types of eucalypt forest are discussed in this manuscript, a wet sclerophyll ecosystem and a dry sclerophyll ecosystem.

14. p11 l22 fig1 E?

We mention two sites in the line identified, so we will include their locations in relation to figure 1, which are 6 for Whroo and 12 for Cumberland Plain.

15. p11 I1-I7 and fig5. Greening ramps of the two ROIs from late nov to late dec,
show approximately a 1 month lag. Could this be related to understory phenological variability? Are the two ROIs looking at the same individuals?

This is another interesting question but the two ROI are not looking at the same individuals. Continued phenocam data collection at this site may help explain this question, but was out of the scope of this study. Fig. 5 is mostly to demonstrate that two different phenocams can identify similar phenology trends, which is why we did not go into extended detail about the lag relationships apparent in the figure.

16. p12 I7-8 here you refer to the site as an "evergreen dry sclerophyll woodland" while in fig 6 caption "temperate eucalypt woodland" is reported. Is it the same? Be more consistent.

They are the same, but as mentioned in a previous comment, we will amend our references throughout the manuscript to be consistent with our terminology.

17. Fig1 and caption. Consider the idea of plotting phenocam site whose date are used in paper (e.g. AU-How, presented in the paper differently form phenocam site not used.

We will make more explicit reference to the phenocams in figure 1 used in the manuscript, to highlight them from the other sites listed.

18. Fig2 Does different green intensity has a meaning?

No, GCC doesn't have any physiological meaning and the magnitude for GCC is a complex combo of "greenness", illumination and camera type and model and setting. Given that this is all referring to camera type and model and setting assuming consistent illumination across the images, the different GCC just "says" that for a given point in time some parts of the image as represented by the various ROI is "greener" than other parts. GCC is simply a relative change, which is why simple and cheap cameras can be used to collect the information (as per Sonnentag et al. 2012).

19. Fig4&6 legend. What does L and R means? It indicates left and right y axes? If

BGD
yes it's not needed.

We will remove the L and R from the figure legend, which should also help address the comment from reviewer 1 about the same issue.

20. Fig4 pics in the lower panel: are these the ROIs used to compute overstory and understory GCC?

Yes, they are examples at least. We will make this clearer in the figure caption.

21. Fig3-5 including ecosystem type in figure caption or plot titles consistent with 4.1 paragraph titles (tropical rainforest, tropical savana, temperate evergreen) will help the reader.

Yes, we will make these ecosystem references more consistent throughout the revised manuscript.

BGD

---

## Editor Comment (EC1) · M. Migliavacca (Editor) · 6 Jul 2016

Dear Authors,

based on the comments posted by the referees and your reply I invite you to submit a properly revised version of the manuscript.

In particular the referees pointed out some abbreviations and acronyms not properly defined and also an improvement of the figures and coherence between figures is needed.

The referee #2 also highlights the need of a more quantitative analysis to compare the EVI trajectories and some works should be done in this direction.

[Figure]

Regarding the answer n 8 to the reviewer 2, I guess the authors refer to daytime averages of the indices and not daily averages. Otherwise the results can be seasonally biased by variations in day-length.

Another important point raised by the referees for the comparison of phenocam and MODIS in Savannahs is the role of fractional cover and understory phenology, and I would add the spatial heterogeneity within the MODIS pixel and representativity of the field of view of the camera. I guess in your sites is not a big issue but more detailed discussion is needed. Looking forward to hearing from you. Best regards